# Cannabidiol Enhances Cabozantinib-Induced Apoptotic Cell Death via Phosphorylation of p53 Regulated by ER Stress in Hepatocellular Carcinoma

**DOI:** 10.3390/cancers15153987

**Published:** 2023-08-05

**Authors:** Youngsic Jeon, Taejung Kim, Hyukjoon Kwon, Jeong-Kook Kim, Young-Tae Park, Jungyeob Ham, Young-Joo Kim

**Affiliations:** 1Natural Products Research Center, Korea Institute of Science and Technology, Gangneung 25451, Republic of Korea; biomangg0@gmail.com (Y.J.); kgsing@kist.re.kr (T.K.); 120053@kist.re.kr (H.K.); pyt1017@kist.re.kr (Y.-T.P.); 2Division of Bio-Medical Science & Technology, KIST School, University of Science and Technology, Seoul 02792, Republic of Korea; 3NeoCannBio Co., Ltd., Seoul 02792, Republic of Korea; jland123@neocannbio.com

**Keywords:** cannabidiol, cabozantinib, apoptosis, p53, endoplasmic reticulum stress

## Abstract

**Simple Summary:**

This study provides evidence that cabozantinib and CBD combination treatment leads to increased apoptosis in HCC cells. We propose that this enhanced apoptosis is mediated, at least in part, through the induction of ER stress and subsequent activation of the p53 pathway. However, our study did not find any association between the combination treatment and CNR.

**Abstract:**

Cannabidiol (CBD), a primary constituent in hemp and cannabis, exerts broad pharmacological effects against various diseases, including cancer. Additionally, cabozantinib, a potent multi-kinase inhibitor, has been approved for treating patients with advanced hepatocellular carcinoma (HCC). Recently, there has been an increase in research on combination therapy using cabozantinib to improve efficacy and safety when treating patients. Here, we investigated the effect of a combination treatment of cabozantinib and CBD on HCC cells. CBD treatment enhanced the sensitivity of HCC cells to cabozantinib-mediated anti-cancer activity by increasing cytotoxicity and apoptosis. Phospho-kinase array analysis demonstrated that the apoptotic effect of the combination treatment was mainly related to p53 phosphorylation regulated by endoplasmic reticulum (ER) stress when compared to other kinases. The inhibition of p53 expression and ER stress suppressed the apoptotic effect of the combination treatment, revealing no changes in the expression of Bax, Bcl-2, cleaved caspase-3, cleaved caspase-8, or cleaved caspase-9. Notably, the effect of the combination treatment was not associated with cannabinoid receptor 1 (CNR1) and the CNR2 signaling pathways. Our findings suggest that the combination therapy of cabozantinib and CBD provides therapeutic efficacy against HCC.

## 1. Introduction

Hepatocellular carcinoma (HCC) is the most common type of primary liver cancer and accounts for over 80% of all primary liver cancer cases worldwide. The incidence of HCC is increasing worldwide, with approximately 800,000 new cases being diagnosed annually [1]. It is a highly aggressive and lethal cancer associated with poor prognosis and limited treatment options. Current treatment options include surgical resection, liver transplantation, chemotherapy, and radiotherapy; however, these treatments have various limitations, including drug resistance, toxicity, and limited efficacy [2]. Therefore, there is a critical need for novel and more effective therapeutic strategies to improve the prognosis and outcomes of patients with HCC.

Presently, combination therapy has emerged as a promising approach for cancer treatment. Combination therapy involves the use of multiple agents that target different pathways associated with tumor growth and metastasis, thereby enhancing treatment efficacy and reducing toxicity [3,4]. The use of protein-targeting agents and natural compounds in combination therapy enhances the anti-tumor effects of individual agents and overcomes drug resistance [5,6,7].

Cabozantinib is a multi-target kinase inhibitor that has shown promising results in clinical trials for treating advanced HCC [8]. Cabozantinib targets several tyrosine kinases, including vascular endothelial growth factor, MET, and tyrosine protein kinase receptor UFO, which are involved in tumor angiogenesis, growth, and metastasis. It improves the overall survival and progression-free survival in patients with HCC who previously received sorafenib, which is the current standard of care for advanced HCC [9,10]. Cannabidiol (CBD), a non-psychoactive component of the *Cannabis sativa*, exhibits anti-tumor activity against various cancers, including HCC [11,12,13]. CBD induces cell cycle arrest, apoptosis, and autophagy, and it inhibits angiogenesis and metastasis in preclinical models of cancer. CBD also enhances the anti-tumor effects of chemotherapeutic agents and helps overcome drug resistance [14,15].

Although interest in the use of CBD combination therapy with anti-cancer drugs is growing, research specifically focusing on cabozantinib combination therapy in HCC cells is limited. Therefore, we aimed to investigate the effects of cabozantinib and CBD combination treatment on HCC cells by assessing cell viability, apoptosis, and potential synergistic interactions.

## 2. Materials and Methods

### 2.1. Information on Origin of Cannabis

The Cheongsam (Korean hemp, *Cannabis sativa* L.) used in this study was cultivated in a licensed facility in the Korea Institute of Science and Technology (KIST) Gangneung Institute (Gangneung-si, Gangwon-do, South Korea). The cultivation and usage of Cheongsam were conducted in compliance with the regulations and guidelines for cannabis research. The assignment/transfer approval processes under drug (cannabis) research permission were obtained from the Ministry of Food and Drug Safety and the Seoul Regional Food and Drug Administration (no. 1564 and 1979).

### 2.2. Purification of Cannabidiol

The Cheongsam leaves were harvested in July 2020, dried, and finely cut. Ethanol extraction (5 L) was performed twice on the naturally dried leaves (500 g) at room temperature, yielding an ethanolic extract (37 g). The extract was then suspended in water and partitioned with normal hexane, resulting in a residue (16 g). Silica open column chromatography (Merck, Darmstadt, Germany; 230–400 mesh, 7.5 × 10.0 cm ID) was conducted using hexane:ethyl acetate eluents in a stepwise fashion. The F2 fraction (4.2 g) was subjected to preparative HPLC (Phenomenex Luna C18 column; 250 × 21.2 mm, 10 µm) eluting with water (A) and MeCN (B), at a flow rate of 10 mL/min via gradient solvent systems (70 to 85% MeCN, 60 min) and using a 220 nm UV detector, which yielded four sub-fractions. Each sub-fraction underwent further purification via semi-preparative HPLC (Phenomenex Luna C18 (2); 250 × 10 mm, 5 µm) eluting with 70~85% MeCN, at a flow rate of 4 mL/min, to obtain pure compounds, including CBD (98 mg), Δ9-THC (12 mg), and Δ9-THCA (15 mg). The F3 fraction (0.8 g) underwent similar purification processes to yield the pure compound CBDA (103 mg). The identification of isolated compounds was confirmed through NMR analysis and comparison with published reports (Appendix A) [16].

### 2.3. Cell Culture and Chemicals

The human HCC cell lines (HepG2 and Hep3B) were obtained from the American Type Culture Collection (ATCC, Manassas, VA, USA). The cells were cultured in MEM (Gibco, Carlsbad, MD, USA) and DMEM (Gibco), respectively, and were supplemented with 10% fetal bovine serum (Gibco), 100 U/mL penicillin, and 100 μg/mL streptomycin. They were maintained at 37 °C in a humidified atmosphere with 5% CO_2_. Cabozantinib was obtained from Sigma-Aldrich (St. Louis, MO, USA) and BML-277 was purchased from Selleck Chemicals (Houston, TX, USA).

### 2.4. Analysis of Cell Viability

Cells (1.0 × 10^4^) were split into 96-well plates and incubated in media for 24 h. Cells were treated with different concentrations of CBD and/or cabozantinib for 48 h, and cell viability was measured using the WST-8 assay kit (Biomax, Seoul, Korea) according to the manufacturer’s recommendations.

### 2.5. Western Blotting

Protein samples were analyzed via Western blotting as described previously [17]. Proteins (whole cell extracts, 30 μg/lane) were separated via electrophoresis in NuPAGE 4–12% Bis-Tris gels (Invitrogen, Carlsbad, CA, USA), blotted onto PVDF transfer membranes, and analyzed using epitope-specific primary and secondary antibodies. Bound antibodies were visualized using SuperSignalTM WestFemto Maximum sensitivity substrate (Thermo Fisher Scientific, San Jose, CA, USA) and an LAS 4000 imaging system (Fujifilm, Japan). The primary and secondary antibodies were used as indicated in Appendix A.

### 2.6. Fluorescence-Activated Cell Sorting (FACS) Analysis

Apoptotic cells were analyzed via flow cytometry using the FITC Annexin apoptosis detection kit (BD Biosciences, CA, USA). HepG2 cells were divided into 4 groups (e.g., control, CBD, CBZ, and combined treatment). After treatment for 48 h, FITC Annexin V and propidium iodide were added to washed cells and incubated for 15 min at room temperature. The apoptotic cells were measured using a FACSVerse Flow Cytometer (BD Biosciences) and data were analyzed using the Flowjo software (version 10.8.1, BD, Ashland, OR, USA). At least 1000 cells were counted for data analysis.

### 2.7. Multi Phospho-Kinase Assay

The multi phospho-kinase activity was measured using a Human Phospho-Kinase array system (R&D System, Minneapolis, NM, USA) according to the manufacturer’s recommendations.

### 2.8. Real-Time PCR Analysis

For mRNA amplification of *GRP78* and *TP53*, quantitative real-time PCR was performed using iQTM SYBR Supermix (Bio-Rad, Hercules, CA, USA). The details of the primer sequences and thermal cycling conditions are summarized in Appendix A.

### 2.9. Dual-Luciferase Reporter Assay

TP53 promoter activity was assessed using a dual-luciferase reporter assay, following a previously described method [18]. Briefly, to investigate the effect on TP53 promoter activity, cells were cotransfected with reporter constructs including pGL3.0-basic (Promega, Madison, WI, USA) and *TP53*-luciferase, along with the pNL1.1.TK vector. After 24 h of transfection, luciferase activity was measured using the Nano-Glo^®^ Dual Luciferase^®^ Reporter Assay System (Promega), following the manufacturer’s protocols. The relative firefly luciferase activity was normalized to NanoLuc™ luciferase activity to account for variations in transfection efficiency.

### 2.10. Public Transcriptome Data Analysis

For transcriptome data analysis, we utilized publicly available data from The Cancer Genome Atlas–Liver Hepatocellular Carcinoma (TCGA-LIHC) dataset, obtained using the TCGAbiolinks package in R software [19]. Additionally, we obtained data from the GSE4024 (accessed on 12 July 2023), GSE89377 (accessed on 12 July 2023), and GSE6764 (accessed on 12 July 2023) datasets from the NCBI GEO database (https://www.ncbi.nlm.nih.gov/geo/, accessed on 12 July 2023). To ensure consistency, samples with normal tissues were excluded from the analysis.

Correlation plots were generated using R software, and gene ontology information was retrieved from the gene ontology database (http://www.geneontology.org/, accessed on 12 July 2023) and reactome database (http://www.reactome.org/, accessed on 12 July 2023). To calculate the expression value of each signature, we computed the mean value of genes associated with the respective signature.

### 2.11. Cloning and Establishment of Overexpression Cell Lines

To clone the tagged *CNR1* and *CNR2* coding gene sequences into the pCDH-CMV-EF1-puro vector, total RNA was extracted from normal liver tissue, followed by the synthesis of cDNA. The CNR1 and CNR2 genes were amplified through PCR using specific primers that contained a 5’-extension and NotI (NEB, Ipswich, MA, USA) and XbaI (NEB) restriction sites. The CloneAmp HiFi PCR Premix (Thermo Fisher Scientific, San Jose, CA, USA) was used for the PCR reaction. Subsequently, the amplicons were obtained via gel extraction from the PCR products and digested with NotI and XbaI enzymes, and then cloned into the pCDH-CMV-EF1-puro vector using the In-Fusion^®^ cloning system according to the manufacturer’s instructions. The primer sequences and thermal cycling conditions are provided in Appendix A.

HepG2 cells were transfected with pCDH-CMV-EF1-puro containing the tagged CNR1 and CNR2 coding sequences, along with the gag-pol and VSV-G plasmids (plasmids 14887 and 8454, respectively; Addgene, Cambridge, MA, USA). The transfection was performed using Lipofectamine^®^ 3000 (Invitrogen, Carlsbad, CA, USA), following the manufacturer’s instructions. After transfection, stable cell lines expressing CNR1 or CNR2 were selected by treating the cells with puromycin (0.5–1.0 µg/mL; Sigma-Aldrich Co., St. Louis, MO, USA) for a period of 4 weeks. To confirm the specificity and efficiency of CNR1 and CNR2 overexpression, mRNA expression was analyzed using RT-PCR.

### 2.12. siRNA-Mediated Knockdown Experiments

p53 (siRNAs; sc-29435, Santa Cruz, CA, USA), c-Jun (siRNAs; sc-29223, Santa Cruz), and MISSION^®^ siRNA Universal Negative Control (Sigma-Aldrich Co.) were transfected into HepG2 cells using Lipofectamine^®^ RNAiMAX Transfection Reagent (Invitrogen), according to the manufacturer’s recommendations. To confirm the specificity and efficiency of p53 and c-Jun inhibition, we analyzed the protein expression levels using Western blotting.

### 2.13. Statistical Analysis

The results are expressed as the means ± S.D. of at least three independent determinations for each experiment. Statistical significance was analyzed using Student’s *t*-tests. All *p*-values were two-tailed, and a *p*-value of less than 0.05 was considered to be significant.

## 3. Results

### 3.1. CBD Did Not Induce Apoptosis in Low-Concentration Treatment

CBD has been previously found to show anti-cancer activity and possess apoptotic properties when used in a high concentration (e.g., >40 μM) against various cancer cells, including breast cancer, lung cancer, and HCC [12,20,21]. To determine the optimal concentration of CBD associated with anti-cancer activity, we first evaluated the half-maximal inhibitory concentration (IC_50_) value of CBD in HepG2 cells, which was found to be 32.52 μM (Figure 1A). Next, we confirmed the apoptotic effects of CBD at different concentrations (0, 3.125, 6.25, 12.5, 25.0, and 50.0 μM) by analyzing the levels of apoptotic marker proteins such as Bcl2-associated X (Bax) and the cleavage of caspase-9, caspase-8, caspase-3, and poly (ADP-ribose) polymerase (PARP). We found a concentration-dependent increase in CBD-induced apoptosis (Figure 1B). Notably, no significant effects were observed at relatively lower concentrations of CBD (<25 μM). Based on these findings, CBD was used at a concentration of 20 μM in further experiments.

### 3.2. CBD Enhanced the Sensitivity to Cabozantinib-Mediated Anti-Cancer Activity

Next, we evaluated the synergistic effects of CBD in combination with various anti-cancer drugs, including sorafenib, cabozantinib, lenvatinib, and regorafenib, on HCC. Among these drugs, cabozantinib combined with CBD (20 μM) demonstrated a higher inhibitory effect than that of other drug combinations, as observed in the inhibition of cell proliferation (IC_50_, 15.76 μM; Figure 2A,B). Additionally, we assessed the synergistic effects of other cannabis compounds, such as cannabidiolic acid (CBDA), CBD, and tetrahydrocannabinol (THC), and found that the combination treatment of cabozantinib and CBD showed a greater inhibitory effect than that of the other combination treatments (IC_50_, 15.76 ± 0.75 μM; Appendix A).

Given that cabozantinib is currently used as a treatment option for progressed HCC [8,9] and that previous studies have highlighted its potential efficiency in inducing apoptosis [22], we investigated whether CBD can enhance the apoptosis-inducing effect of cabozantinib. The combination treatment of cabozantinib and CBD increased the levels of apoptotic protein markers, including Bax, cleaved caspase-9, cleaved caspase-8, cleaved caspase-3, and cleaved PARP, in a dose-dependent manner when compared to the CBD-only treatment. Conversely, the expression of Bcl-2, an inhibitor of apoptotic stimuli, showed a decreasing pattern (Figure 2C). To analyze the apoptotic rate, we performed fluorescence-activated cell sorting (FACS) and observed that the proportions of early and late apoptotic cells were higher in the cabozantinib and CBD co-treated HCC cells than in those treated with CBD alone (Figure 2D,E). Therefore, CBD enhanced the sensitivity of HCC cells to cabozantinib-induced apoptosis.

### 3.3. Synergistic Effects of the Combination Treatment of Cabozantinib and CBD Were Associated with Chk2 and p53

To elucidate the underlying mechanisms of the combination treatment of cabozantinib and CBD, we performed a multi phospho-kinase assay to identify the signaling pathways modulated by the combination treatment. Using a multi phospho-kinase assay, we screened putative kinase proteins involved in cell proliferation, differentiation, cell cycle regulation, and apoptosis. Notably, a significant enrichment was found in the levels of phosphorylated checkpoint kinase 2 (Chk2), c-Jun, and p53 after the combination treatment, compared to that obtained using the single treatments (Figure 3A). These proteins are associated with apoptosis [23,24,25]. Furthermore, Chk2 activation induces apoptosis and cell cycle arrest via p53 phosphorylation [25]. Additionally, we observed an increase in the levels of phosphorylated c-Jun, Chk-2, and p53 induced by the combination treatment over time, compared to that obtained using the cabozantinib-only treatment (Figure 3B).

To verify whether Chk2 activation can affect apoptosis following the combination treatment of cabozantinib and CBD, we examined the Chk2-inhibitor-mediated anti-apoptotic effects. The treatment with BML-277, an ATP-competitive inhibitor of Chk2, suppressed the apoptotic effects of the cabozantinib and CBD combination treatment (Figure 3C). In addition, we evaluated the inhibitory effects of BML-277 treatment with cabozantinib and CBD combination treatment on Chk2 and p53 phosphorylation and apoptotic protein levels; BML-277-mediated Chk2 inhibition decreased the phosphorylation of p53 and levels of apoptotic proteins (Figure 3D).

Next, to elucidate the underlying mechanisms of the combination treatment in apoptosis, we evaluated apoptosis based on the expression of p53. Using the small interfering RNA (siRNA) system, HCC cells exhibiting p53 gene knockdown (transfected with si*TP53*) demonstrated increased cell survival compared to that of cells transfected with siCtrl when treated with cabozantinib in a dose-dependent manner (siCtrl vs. si*TP53*, *p* < 0.01; Figure 3E). Furthermore, Hep3B cells, the p53-null HCC cell line, exhibited a lower response to the cabozantinib and CBD combination treatment than that of HepG2 cells (IC_50_, 32.1 μM; Appendix A and Figure 2A). Given that p53 has been implicated in the regulation of cell apoptosis [23,26], we further evaluated the relationship between apoptotic markers (cleaved PARP, cleaved caspase-9, caspase-8, and caspase-3) and p53 activation mediated by the cabozantinib and CBD combination treatment. si*TP53*-transfected HCC cells displayed significantly decreased levels of cleaved PARP, caspase-9, caspase-8, and caspase-3 compared to those in siCtrl-transfected HCC cells (Figure 3F). The effect of *JUN* knock-down on apoptosis after the combination treatment was evaluated; however, no significant difference was observed (Appendix A). Taken together, it is plausible that apoptosis following the combination treatment of cabonzantinib and CBD may be associated with p53 activation.

### 3.4. Cabozantinib and CBD Combination Treatment Enhanced TP53 Activation via ER Stress

Based on the previously determined association between p53 activation and p53-dependent apoptosis with endoplasmic reticulum (ER) stress [27,28], our initial investigation attempted to determine whether p53 activation was associated with ER stress. To explore this, we analyzed human transcriptome datasets, including TCGA-LIHC, GSE4024, GSE89377, and GSE6764. Our analysis revealed a significant correlation between the expression levels of p53-activation-related genes and ER-stress-related genes in independent cohorts, such as TCGA-LIHC, GSE4024, and GSE6764 (Figure 4A). Furthermore, we examined the effects of cabozantinib and CBD combination treatment on the expression of glucose-regulated protein 78 (GRP78), an ER stress marker. Notably, the combination treatment increased *GRP78* expression compared to that obtained using individual treatments of cabozantinib and CBD, revealing the increased *TP53* gene expression and transcription (Figure 4B,C). Next, to monitor cellular ER stress, we assessed the levels of phosphorylated protein kinase RNA-like endoplasmic reticulum kinase (PERK) and eukaryotic initiation factor 2 alpha (eIF2α) and observed that the combination treatment increased the levels of phosphorylated proteins compared to those obtained using single treatment with cabozantinib alone, according to time (Figure 4D). Previous studies have shown that L-ascorbic acid (LAA) can ameliorate ER stress [29,30]; thus, we investigated whether LAA could inhibit ER-stress-mediated p53 activation and apoptosis. LAA treatment suppressed ER-stress-induced p53 activation and cell apoptosis (Figure 4E–G).

### 3.5. Synergistic Effects of Cabozantinib and CBD Combination Treatment were Not Associated with Cannabinoid Receptors

CBD has been shown to bind to cannabinoid receptor 1 (CNR1) and CNR2 and regulate their downstream molecular pathways [31]. Therefore, we aimed to identify the key downstream molecules involved in the synergistic apoptotic effects of CBD in combination with cabozantinib. We initially examined whether the expression of CNR1 and CNR2, which are major receptors coupled to CBD, can influence these effects. Therefore, we established a CNR overexpression system in HepG2 cells, which showed lower endogenous expression levels of *CNR1* and *CNR2* (Figure 5A). Our findings demonstrated that the synergistic effects, including cell proliferation, observed after the cabozantinib and CBD combination treatment were not associated with CNR1 or CNR2 (Figure 5B,C).

Next, we performed FACS analysis to assess whether the expression of CNR1 and CNR2 can regulate the apoptotic effects of the cabozantinib and CBD combination treatment. Notably, the cells expressing CNR1 and CNR2 exhibited a similar population of apoptotic cells, including both early and late apoptotic cells, compared to that in the control cells treated with cabozantinib and CBD (Figure 5D and Appendix A). Additionally, Western blot analysis confirmed that control cells treated with cabozantinib and CBD showed comparable expression levels of apoptotic proteins, such as cleaved PARP, cleaved caspase-9, cleaved caspase-8, and Bax, to those in cells expressing CNR1 and CNR2 (Figure 5E). Taken together, the synergistic apoptotic effects mediated by cabozantinib and CBD combination treatment were not dependent on CNR1 or CNR2.

## 4. Discussion

In this study, we investigated the potential synergistic effects of cabozantinib and CBD co-treatment on HCC cells. We found that CBD enhanced the sensitivity of HCC cells to cabozantinib-mediated inhibition, resulting in increased cytotoxicity and apoptosis. Notably, we found that p53 phosphorylation, which is regulated by ER stress, serves as a significant mediator of the apoptotic effect observed in the combination treatment, specifically when compared to effects obtained via treatment with one drug alone.

Previous studies have highlighted the anti-cancer effects of CBD, including its apoptotic properties, on various cancer cells, including breast cancer, lung cancer, and HCC [12,20,21]. Similarly, we observed a concentration-dependent increase in CBD-induced apoptosis in HepG2 cells. However, it should be noted that CBD treatment induced apoptotic cell death at relatively higher concentrations in this study. Therefore, there is evidence that CBD use alone is insufficient for treating HCC.

The rationale for this study was based on the pharmacological properties of CBD, including its broad potential in treating diseases, including cancer. Additionally, cabozantinib has been approved for use in patients with advanced HCC, leading to an increasing interest in the use of combination therapy to improve treatment efficacy and safety [9,32]. Our results provide compelling evidence that supports the potential benefits of combining cabozantinib with CBD for HCC treatment.

To evaluate the synergistic effects of CBD and different anti-HCC drugs, we tested combinations of CBD with sorafenib, cabozantinib, lenvatinib, and regorafenib. Among these drugs, cabozantinib showed the strongest inhibitory effect against cell proliferation when used in combination with CBD. We also compared the synergistic effects of other cannabis compounds, such as CBDA, CBD, and THC, and found that the combination of cabozantinib and CBD exhibited the most significant anti-cancer effect.

One of the key findings of our study was p53 activation after combination treatment, determined using multi-phospho-kinase data analysis and cell culture experiments, and we propose that this activation is mediated via the induction of ER stress. Combination treatment increased the levels of ER stress markers such as *GRP78*. This suggests that the combination treatment induces ER stress in HCC cells. In addition, we demonstrated that the combination treatment enhanced Chk2 activation, revealing the phosphorylation and activation of p53. Activated p53 can subsequently promote apoptosis via the upregulation of apoptotic proteins, such as Bax, cleaved caspase-9, cleaved caspase-8, and cleaved caspase-3, and via the downregulation of the anti-apoptotic factor Bcl-2.

Notably, our study did not find any association between the combination treatment and CNRs. While CBD is known to interact with CNRs [31], the effects observed in our study can be attributed to the ER-stress-mediated Chk2-p53 pathway, rather than direct interactions with CNRs. Further research is required to fully elucidate the precise molecular mechanisms underlying the enhanced apoptosis observed after the combination treatment of cabozantinib and CBD. Additionally, in vivo studies and clinical trials are required to validate the potential therapeutic benefits of this combination approach in patients with HCC.

## 5. Conclusions

In this study, we aimed to investigate the effects of the combination treatment of cabozantinib and CBD on HCC cells and explore the underlying mechanisms of its anti-tumor activity. Specifically, we focused on the apoptotic cell death pathway, which is a critical mechanism of cancer cell death. We hypothesized that the combination of cabozantinib and CBD will enhance apoptotic cell death in HCC cells via p53 phosphorylation, which is regulated by ER stress. These findings support the potential of the combined use of these two agents as a therapeutic strategy for HCC and warrant further investigation for clinical translation.

## Figures and Tables

**Figure 1 cancers-15-03987-f001:**
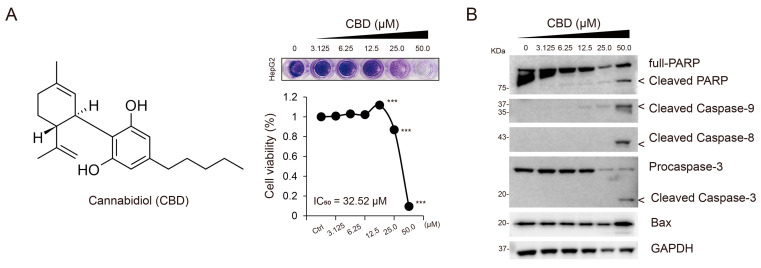
CBD does not induce apoptosis in low-concentration treatment. (**A**) Structure of CBD (**left**). Cytotoxicity mediated by CBD was assessed via crystal violet staining and WST-8 assay at different concentrations (0, 3.125, 6.25, 12.5, 25, and 50 μM; (**right**)). Statistical significance is indicated based on *** *p*  <  0.001 (Student’s *t*-test). (**B**) The levels of apoptosis-related proteins including PARP, cleaved caspase-9, cleaved caspase-8, cleaved caspase-3, and Bax were assessed using Western blot analysis. Protein levels were normalized to those of GAPDH. CBD, cannabidiol; PARP, poly (ADP-ribose) polymerase; GAPDH, glyceraldehyde 3-phosphate dehydrogenase.

**Figure 2 cancers-15-03987-f002:**
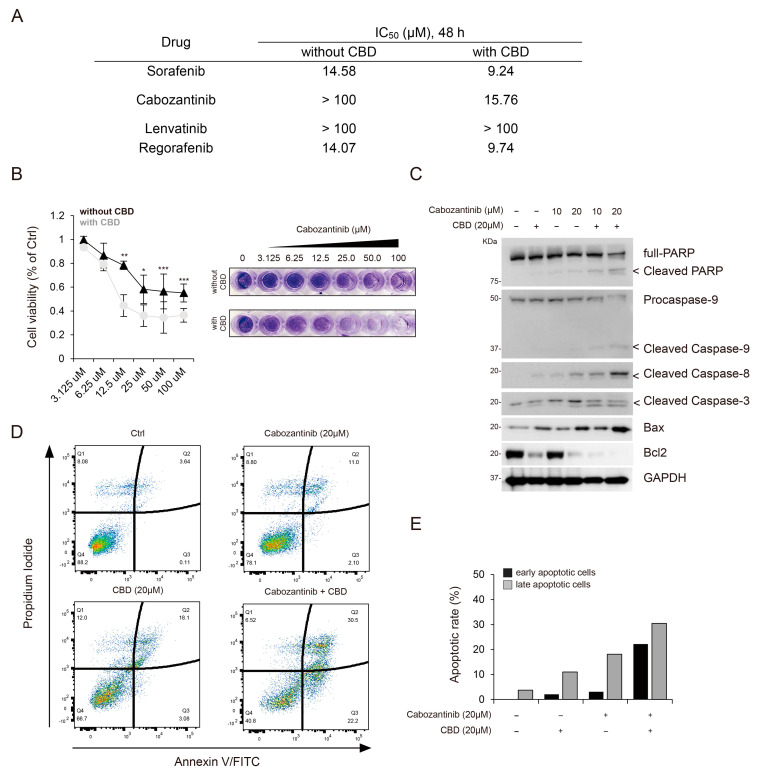
CBD enhances sensitivity to cabozantinib-mediated anti-cancer activity. (**A**) The IC_50_ values of sorafenib, cabozantinib, lenvatinib, and regorafenib with and without CBD. (**B**) Cytotoxicity mediated by cabozantinib with and without CBD was assessed using the WST-8 assay (**left**) and crystal violet staining (**right**) at different concentrations. Statistical significance is indicated (without CBD vs. with CBD; * *p*  <  0.05, ** *p*  <  0.01, and *** *p*  <  0.001; Student’s *t*-test). (**C**) The level of apoptosis-related proteins (PARP, cleaved caspase-9, cleaved caspase-8, cleaved caspase-3, and Bax) was assessed by performing Western blotting. Protein levels were normalized to those of GAPDH. (**D**,**E**) Fluorescence-activated cell sorting analysis of early apoptosis in HepG2 cells incubated in the indicated condition. Early apoptotic cells were identified based on the increase in fluorescence intensity of FITC-labeled Annexin V. IC_50_, half-maximal inhibitory concentration; CBD, cannabidiol; PARP, poly (ADP-ribose) polymerase; GAPDH, glyceraldehyde 3-phosphate dehydrogenase.

**Figure 3 cancers-15-03987-f003:**
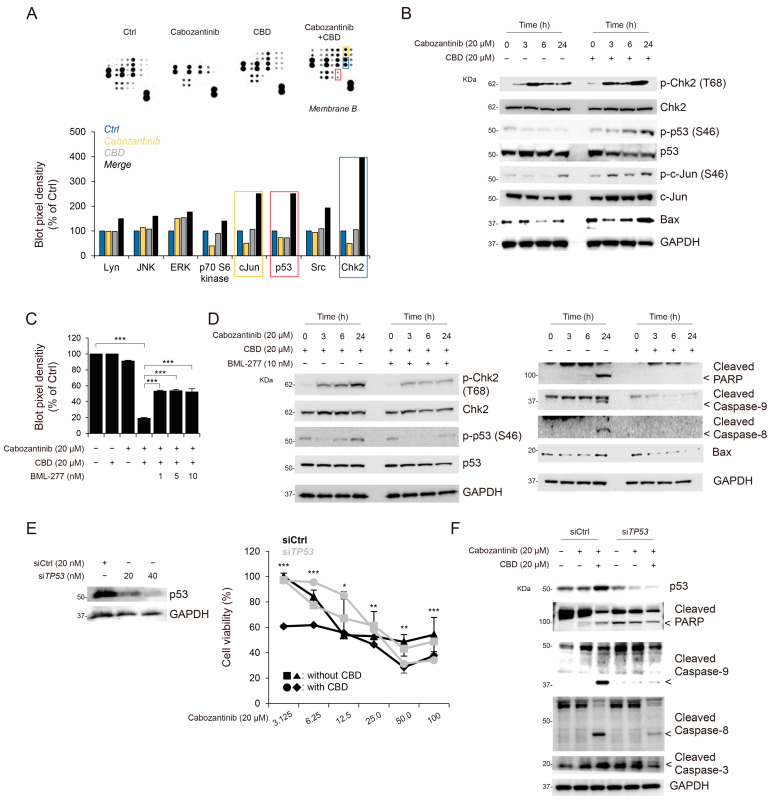
Synergistic effects of cabozantinib and CBD combination treatment are associated with Chk2 and p53. (**A**) Representative images showing a selected region of phospho-kinase array. The levels of phosphorylated Chk2, c-Jun, and p53 on individual membranes represented by the array compared with control cells (**top**). Quantitative analysis of the spots was performed via densitometry and presented as a fold change (**bottom**). (**B**) HepG2 cells were treated with cabozantinib with and without CBD for the indicated time (0, 3, 6, and 24 h). Phospho-Chk2, Chk-2, Phospho-p53, p53, Phospho-c-Jun, c-Jun, and Bax levels were determined using Western blot analysis. GAPDH was used as a control marker. (**C**) Bar plot showing cell viability. Statistical significance is indicated (* *p*  <  0.05, ** *p*  <  0.01, and *** *p*  <  0.001; Student’s *t*-test). (**D**) HepG2 was treated with the indicated agents for the indicated time (0, 3, 6, and 24 h). Phospho-Chk2, Chk-2, Phospho-p53, p53, cleaved caspase-9, cleaved caspase-8, and cleaved caspase-3 levels were determined using Western blot analysis. GAPDH was used as a control marker. (**E**) *TP53* siRNA (20 nM) and control siRNA (20 nM) were used to transfect HepG2 cells for 48 h; cytotoxicity induced by cabozantinib with and without CBD was assessed using WST-8 assay (**left**) and Western blotting (**right**) at different concentrations. (**F**) TP53 siRNA (20 nM) and control siRNA (20 nM) were used to transfect HepG2 cells, and the levels of apoptosis-related proteins (PARP, cleaved caspase-9, cleaved caspase-8, cleaved caspase-3, and Bax) were assessed using Western blotting. Protein levels were normalized to those of GAPDH. CBD, cannabidiol; Chk2, checkpoint kinase 2; siRNA, small interfering RNA; PARP, poly (ADP-ribose) polymerase; GAPDH, glyceraldehyde 3-phosphate dehydrogenase.

**Figure 4 cancers-15-03987-f004:**
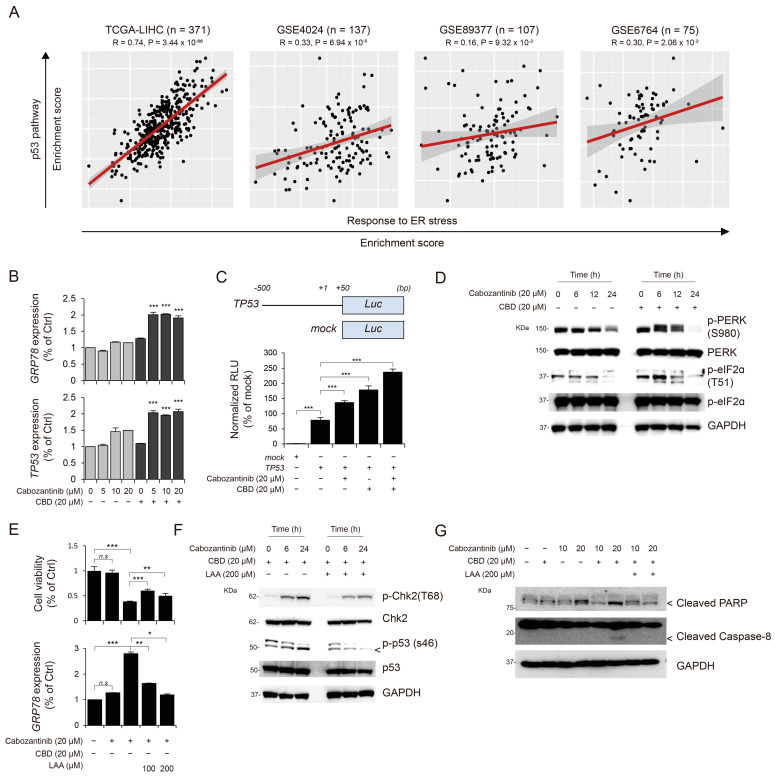
Cabozantinib combined with CBD enhances p53 activation via ER stress. (**A**) Correlations between the expression of TP53-pathway-related genes and the expression of ER-stress-related genes in the independent cohort (*n* = 4) are shown (for details, see Section 2). (**B**) Bar plots showing expression levels of *GRP78* (**top**) and *TP53* (**bottom**) after treatment with cabozantinib and with and without CBD. Statistical significance is indicated (without CBD vs. with CBD; *** *p*  <  0.001; Student’s *t*-test). (**C**) *TP53* promoter region (−500 to +50 from the transcription start site [+1]) and protein expression were evaluated using the luciferase activity assay in HepG2 cells. (**D**) The level of ER-stress-related proteins (phosphor-PERK, PERK, phospho-eIF2α, and eIF2α) was determined using Western blotting. Protein levels were normalized to those of GAPDH. (**E**) Bar plots showing cell viability and *GRP78* expression. Statistical significance is indicated (* *p*  <  0.05, ** *p*  <  0.01, and *** *p*  <  0.001; Student’s *t*-test). (**F**,**G**) HepG2 cells were treated with the indicated treatment for the indicated time (0, 12, and 24 h). Phospho-Chk2, Chk-2, Phospho-p53, p53, cleaved PARP, and cleaved caspase-8 levels were determined using Western blot analysis. GAPDH was used as a control marker. ER, endoplasmic reticulum; CBD, cannabidiol; Chk2, checkpoint kinase 2; PARP, poly (ADP-ribose) polymerase; GAPDH, glyceraldehyde 3-phosphate dehydrogenase; PERK, protein kinase RNA-like endoplasmic reticulum kinase; eIF2α, eukaryotic initiation factor 2 alpha.

**Figure 5 cancers-15-03987-f005:**
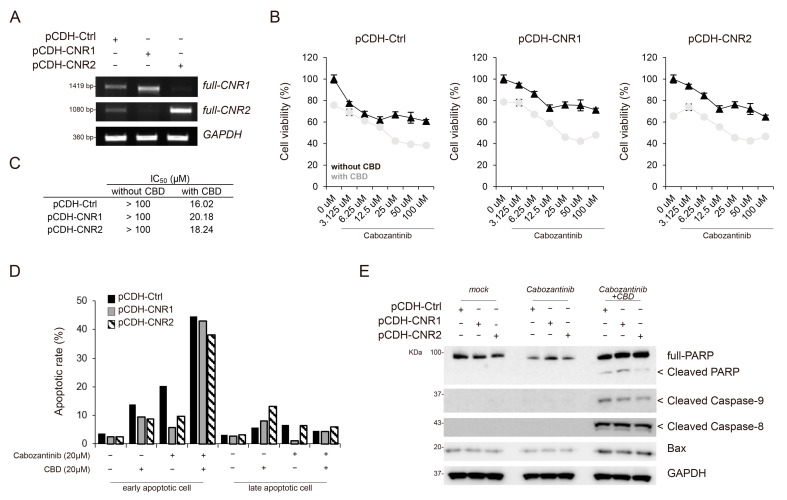
Synergistic effects of cabozantinib and CBD combined treatment are not associated with cannabinoid receptors. (**A**) The expression levels of *CNR1* and *CNR2* in stable cell lines were determined using reverse transcription polymerase chain reaction (for details, see Section 2). The levels were normalized to those of *GAPDH*. (**B**) Cytotoxicity induced by cabozantinib with and without CBD in each stable cell line was assessed using WST-8 assay at diverse concentrations. (**C**) The IC_50_ values of cabozantinib with and without CBD for each stable cell line. (**D**) Bar plot showing apoptosis rate based on fluorescence-activated cell sorting. (**E**) The expression level of apoptosis-related proteins (PARP, cleaved caspase-9, cleaved caspase-8, cleaved caspase-3, and Bax) in each stable cell line was assessed using Western blotting. The levels were normalized to those of GAPDH. IC_50_, half-maximal inhibitory concentration; CNR1, cannabinoid receptor 1; CBD, cannabidiol; PARP, poly (ADP-ribose) polymerase; GAPDH, glyceraldehyde 3-phosphate dehydrogenase.

## Data Availability

The data generated and analyzed during the current study are available from the corresponding author upon reasonable request.

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
