# Peer review of "Cannabidiol Enhances Cabozantinib-Induced Apoptotic Cell Death via Phosphorylation of p53 Regulated by ER Stress in Hepatocellular Carcinoma"

_cancers, 2023, doi:10.3390/cancers15153987_

Round 1

Reviewer 1 Report

1.     Please provide a citation that supports this categorical statement. ‘’The use of protein-targeting agents and natural compounds in combination therapy enhances the antitumor effects of the individual agents and overcomes drug resistance’’

2.     Please provide the meaning of all abbreviations at first use in the text

3.     2.5. Western blotting      

It is recommended that the authors provide a detailed description of the western blotting procedures used.

4.     Due to the fact that Bcl-2 can occasionally inhibit and sometimes induce apoptosis, I am curious as to why XIAP was not chosen instead of Bcl-2 by the authors

5.     The use of CBD is associated with adverse drug interactions, liver toxicity, and reproductive and developmental effects. Due to the high dose of CBD herein, the combination with cabozantinib, which is toxic, is quite alarming. Is this feasible for humans?

6.     A study of the combined use of CBD and cabozantinib in vivo is needed to assess its effects on HCC. In addition, the combination of CBD and cabozantinib should be tested for toxicity.

Author Response

Response to Reviewer 1 comments

Comments and suggestions for authors

Point 1: Please provide a citation that supports this categorical statement. ‘’The use of protein-targeting agents and natural compounds in combination therapy enhances the antitumor effects of the individual agents and overcomes drug resistance’’

Response 1: We have added reference to Introduction section.

Point 2: Please provide the meaning of all abbreviations at first use in the text

Response 2: We have added the abbreviation in the main text on page 2.

Point 3: 2.5. Western blotting It is recommended that the authors provide a detailed description of the western blotting procedures used.

Response 3: According to your advice, we have revised the manuscript in the materials and methods section on page 5 accordingly.

Point 4: Due to the fact that Bcl-2 can occasionally inhibit and sometimes induce apoptosis, I am curious as to why XIAP was not chosen instead of Bcl-2 by the authors

Response 4: It is known that the life-or-death decision for a cell is mainly determined by the interactions between three factions of the BCL-2 family: namely, the pro-survival subfamily (e.g., BCL-2, BCL-XL and MCL1) and two pro-apoptotic factions, the BH3-only proteins (e.g., BIM, PUMA and BID), which convey various cytotoxic signals, and the death effectors BAX and BAK, which can convert into homo-oligomers that perforate the mitochondrial outer membrane, triggering the proteolytic cascade that demolishes the cell (1). In simple terms, Bcl-2, Bcl-XL and MCL1 are anti-apoptotic factors, and BAX and BAK are pro-apoptotic factors. Therefore, we chose caspases in addition to BCL-2 and BAX, whose expression patterns were found to be opposite, to confirm the apoptotic induction of HCC cells through combination therapy. If, as you suggested, we had additionally examined the expression pattern of XIAP, we might have obtained even better results.

Point 5: The use of CBD is associated with adverse drug interactions, liver toxicity, and reproductive and developmental effects. Due to the high dose of CBD herein, the combination with cabozantinib, which is toxic, is quite alarming. Is this feasible for humans?

Response 5: Recently, CBD has been increasingly used in various industries, including pharmaceuticals, cosmetics, food, and beverages, in many countries. Despite concerns about its safety and efficacy, the popularity of CBD continues to grow. However, the lack of sufficient information on CBD drug interactions underscores the importance of evidence-based research and knowledge. In this study, we treated HCC cells with CBD at a non-toxic low dose. For potential future human applications, further research on dosage and other factors is necessary. Nevertheless, our research findings indicate that the combined treatment of cabozantinib with CBD shows promising synergistic anticancer activity for HCC treatment.

Point 6. A study of the combined use of CBD and cabozantinib in vivo is needed to assess its effects on HCC. In addition, the combination of CBD and cabozantinib should be tested for toxicity.

Response 6: I wholeheartedly agree with your suggestion. However, for this study, our priority was to promptly report the improved anti-cancer effects of combination therapy with CBD and cabozantinib. We kindly ask for your understanding in this regard. Going forward, we intend to conduct animal experiments, including toxicity testing, based on the results of this study.

Reviewer 2 Report

Dear author, 

This work is nicely done with great target validation and controls. The proof that the CBD enhanced HCC sensitivity of cabozantinib is solid. In addition, the observation of p53 phosphorylation is true that it was upregulated by CBD.

The author also successfully showed that the p-p53 upregulation was the outcome from the combination treatment-induced ER stress with the ER stress biomarker. 

Overall, no significant flaws and obvious errors were found. 

1. What is the main question addressed by the research? The research has found the CBD could enhance the HCC sensitivity from combination therapy. The author found that the CBD could upregulate the p-p53 and induce cell stress for the HCC apoptosis.   2. Do you consider the topic original or relevant in the field, and if so, why? The observation is original and the proof of the concept data is convincing.   3. What does it add to the subject area compared with other published material?   4. What specific improvements could the authors consider regarding the methodology? There are many more experiments the authors could do, such as the RNA-seq, proteomic analysis, and gene knockout. They could also do the pulldown of the CBD and validate the actual cellular target to understand the true mechanism of actions of the CBD. But for this paper, I believe the experiments are sufficient.   5. Are the conclusions consistent with the evidence and arguments presented and do they address the main question posed? Yes the conclusion has successfully summarized the paper content and address the reasons for HCC sensitivity increase after CBD combination.   6. Are the references appropriate?

Yes I reviewed the reference and it should be good.

Author Response

Response to Reviewer 2 comments

Comments and suggestions for authors

This work is nicely done with great target validation and controls. The proof that the CBD enhanced HCC sensitivity of cabozantinib is solid. In addition, the observation of p53 phosphorylation is true that it was upregulated by CBD.

The author also successfully showed that the p-p53 upregulation was the outcome from the combination treatment-induced ER stress with the ER stress biomarker.

Overall, no significant flaws and obvious errors were found.

Point 1: What is the main question addressed by the research? The research has found the CBD could enhance the HCC sensitivity from combination therapy. The author found that the CBD could upregulate the p-p53 and induce cell stress for the HCC apoptosis.

Response 1: As you mentioned, the main objective of this study is to investigate the potential synergistic effect of cannabis-derived natural compounds, including CBD, when combined with anti-cancer drugs used for HCC. Among these drugs, cabozantinib was selected for investigation. Our study findings indicate that the combination treatment of cabozantinib and CBD could upregulate p-p53 and induce cell stress, leading to apoptosis in HCC cells.

Point 2: Do you consider the topic original or relevant in the field, and if so, why? The observation is original and the proof of the concept data is convincing.

Answer:

Response 2: Yes, the topic is considered original and relevant in the field. The observations presented in the study are novel, and the proof of concept data provided is convincing. The investigation of the potential synergistic effect of cannabis-derived natural compounds, specifically CBD, in combination with anti-cancer drugs (e.g., cabozantinib) for hepatocellular carcinoma (HCC) is a significant and valuable contribution to the field of cancer research. The findings offer new insights into the potential therapeutic benefits of combining CBD with cabozantinib for the treatment of HCC, opening up avenues for further research and clinical applications.

Point 3: What does it add to the subject area compared with other published material?

Response 3: This study adds value to the subject area by investigating the combination effect of CBD and anti-cancer drugs (e.g., cabozantinib) for HCC. Its original observations and convincing data contribute to the growing knowledge on CBD's therapeutic potential with cabozantinib for HCC treatment.

Point 4: What specific improvements could the authors consider regarding the methodology? There are many more experiments the authors could do, such as the RNA-seq, proteomic analysis, and gene knockout. They could also do the pulldown of the CBD and validate the actual cellular target to understand the true mechanism of actions of the CBD. But for this paper, I believe the experiments are sufficient.

Response 4: Consistent with the reviewer's perspective, while the manuscript presents sufficient experiments, these additional investigations could strengthen the study's findings and contribute to a more in-depth understanding of the synergistic effect between CBD and cabozantinib for HCC treatment. Thank you for your advice.

Point 5: Are the conclusions consistent with the evidence and arguments presented and do they address the main question posed? Yes the conclusion has successfully summarized the paper content and address the reasons for HCC sensitivity increase after CBD combination.

Response 5: Thank you for the positive feedback.

Point 6: Are the references appropriate?

Response 6: We have incorporated additional references in the Introduction section, and we are confident that the other references cited in the paper are relevant and appropriate.
